# Assessing the Role of MicroRNAs in Predicting Breast Cancer Recurrence—A Systematic Review

**DOI:** 10.3390/ijms24087115

**Published:** 2023-04-12

**Authors:** Luis Bouz Mkabaah, Matthew G. Davey, James C. Lennon, Ghada Bouz, Nicola Miller, Michael J. Kerin

**Affiliations:** 1Discipline of Surgery, Lambe Institute for Translational Research, University of Galway, H91 YR71 Galway, Ireland; l.bouzmkabaah1@nuigalway.ie (L.B.M.); j.lennon1@nuigalway.ie (J.C.L.);; 2Faculty of Pharmacy in Hradec Králové, Charles University, 50005 Hradec Králové, Czech Republic

**Keywords:** breast cancer, microRNAs, recurrence, survival outcomes, oncological outcomes

## Abstract

Identifying patients likely to develop breast cancer recurrence remains a challenge. Thus, the discovery of biomarkers capable of diagnosing recurrence is of the utmost importance. MiRNAs are small, non-coding RNA molecules which are known to regulate genetic expression and have previously demonstrated relevance as biomarkers in malignancy. To perform a systematic review evaluating the role of miRNAs in predicting breast cancer recurrence. A formal systematic search of PubMed, Scopus, Web of Science, and Cochrane databases was performed. This search was performed according to the Preferred Reporting Items for Systematic Review and Meta-Analyses (PRISMA) checklist. A total of 19 studies involving 2287 patients were included. These studies identified 44 miRNAs which predicted breast cancer recurrence. Results from nine studies assessed miRNAs in tumour tissues (47.4%), eight studies included circulating miRNAs (42.1%), and two studies assessed both tumour and circulating miRNAs (10.5%). Increased expression of 25 miRNAs were identified in patients who developed recurrence, and decreased expression of 14 miRNAs. Interestingly, five miRNAs (miR-17-5p, miR-93-5p, miR-130a-3p, miR-155, and miR-375) had discordant expression levels, with previous studies indicating both increased and reduced expression levels of these biomarkers predicting recurrence. MiRNA expression patterns have the ability to predict breast cancer recurrence. These findings may be used in future translational research studies to identify patients with breast cancer recurrence to improve oncological and survival outcomes for our prospective patients.

## 1. Introduction

Breast cancer is the most common malignancy diagnosed in women and the second leading cause of cancer-related mortality [1]. Survival outcomes correlate directly with the stage at the time of diagnosis, with earlier detection conferring favourable oncological and survival outcomes. Furthermore, disease recurrence is associated with poorer outcomes [2,3], and the extent of such recurrences has been illustrated to impact survival [4]. Notwithstanding, identifying patients at risk of disease recurrence poses a clinical challenge. Accordingly, discovering accurate, reproducible, and minimally invasive biomarkers with the potential to aid the early detection of disease recurrence is essential in improving oncological and survivorship outcomes for those with previous breast cancer diagnoses.

MicroRNAs (miRNAs) are small non-coding RNAs which are approximately 19 to 25 nucleotides in length [5] which are now recognised as key modulators of genetic expression [6]. MicroRNAs act at a post-transcriptional level by binding to messenger RNA (mRNA) and influencing protein synthesis at a cellular level [7]. Recent data has illustrated the relevance of miRNA measurement to predict response to therapeutic strategies [7,8,9,10], enhancing cancer diagnostics [11,12], and predicting long-term outcomes for those diagnosed with malignancy [13,14]. In combination, these favourable characteristics illustrate the candidacy of miRNAs as biomarkers in the context of breast cancer research [15,16,17,18].

Contemporary research supports the hypothesis that miRNAs have clinical utility as biomarkers capable of estimating prognosis in early-stage primary breast cancer [19]. Furthermore, the literature suggests that miRNAs may be responsible for the dormancy and latent metastases hypothesis perpetuated in breast cancer recurrence [18]. Differentially expressed circulating (ct-miRNAs) and tumour-based miRNAs in patients with early-stage breast cancer may predict relapse prior to symptoms of recurrence developing as they influence defective behaviour, relapse risk, and ultimately survival [19]. Accordingly, the aim of this systematic review was to identify a panel of miRNAs which may predict disease recurrence in patients diagnosed with early breast cancer.

## 2. Methods

### 2.1. Literature Search

A formal systematic search was performed of the PUBMED, SCOPUS, Web of Science, and COCHRANE databases in accordance with the Preferred Reporting Items for Systematic Review and Meta-Analyses (PRISMA) checklist [20]. An initial predefined search strategy was outlined by the senior authors (M.G.D, N.M., and M.J.K.). Two authors (L.B.M and J.L.) conducted independent searches of the four databases to identify studies suitable for inclusion The latest of which occurred in September 2022. Discrepancies in opinion were arbitrated by a third author (M.G.D.). The search terms were as follows: (microrna), (mirna), (breast cancer), (recurrence) which were all linked with the Boolean operators ‘AND’ and ‘OR’. Studies were combined from the four databases and duplicate studies were then manually removed. Studies were limited to those published in the English language. Studies were not restricted based on year of publication. Proceedings from academic conferences were excluded. All studies had their titles screened initially and studies considered to be relevant had their abstracts and full texts reviewed. The Quality Assessment of Diagnostic Accuracy Studies-2 (Quadas-2) was used to determine the diagnostic quality of included miRNAs [21].

### 2.2. Eligibility Criteria

Studies meeting the following predetermined inclusion criteria were considered for inclusion in this systematic review: studies assessing the role of miRNAs in predicting breast cancer recurrence in human tissue (i.e., blood, plasma, serum, or tumour tissue). Studies failing to meet these criteria were excluded (this included studies using animal models or cell lines). Eligible studies had to provide full-text manuscripts. Review articles, conference abstracts, and case reports were excluded.

The following details were extracted from studies meeting this eligibility criteria: (1) name of the first author, (2) year of publication, (3) tissue used in evaluation, (4) methodology and laboratory techniques, (5) treatment received, whether surgical or neoadjuvant or adjuvant chemoradiotherapy miRNA appraised, (6) miRNA appraised. Studies reporting data from the same translational research facility were evaluated for the duplication of patient data and studies with overlapping patient data were excluded.

### 2.3. Statistical Analysis

Clinicopathological data were presented as proportions using basic descriptive statistics. MiRNA expression levels and treatment characteristics were described using narrative statistics.

## 3. Results

### 3.1. Literature Search

The literature search yielded a total of 1847 studies. Following the removal of 587 duplicate studies, 1260 independent studies had their titles screened for relevance for inclusion in this study. Of these, 380 had their abstracts reviewed for relevance and 132 required full-text manuscript review. Overall, nineteen studies were included in this systematic review, of which one was a prospective, multicentre clinical trial [14]. All of the other 18 included studies were of retrospective design [22,23,24,25,26,27,28,29,30,31,32,33,34,35,36,37,38,39]. A PRISMA flow diagram detailing the systematic search process is outlined in Figure 1.

### 3.2. MiRNA Expression Profiling Methodology

Thirteen studies (68.4%) used a microarray technique to select suitable miRNA for analysis [22,24,26,29,30,33,34,35,36,37,38,39], five studies (21.1%) used literature review [14,23,25,27,31], and two studies (10.5%) selected miRNAs from historical lab findings [28,32]. All 19 studies (100%) used quantitative reverse transcription polymerase chain reaction (qRT-PCR) for assessing miRNAs [14,22,23,24,25,26,27,28,29,30,31,32,33,34,35,36,37,38,39] (Table 1).

### 3.3. Disease Recurrence

In total, 34.7% of patients experienced disease recurrence at a mean follow up of 94.9 months (793/2287—18 studies). The extent of recurrence was reported in 143 patients, and of these, 54 patients developed locoregional recurrence (LRR) (6.8%) and 89 patients developed distant recurrence (DR) (11.2%) (5 studies). Pathological confirmation of primary and recurrent breast cancer was specifically included in 15 studies [14,22,23,25,26,27,28,29,31,32,35,36,37,38,39]. Basic treatment characteristics for each study are outlined in Table 1.

**Table 1 ijms-24-07115-t001:** Included studies assessing miRNA expression and their role in breast cancer recurrence.

Author	Year	Country	Tissue	N	LOE	Pathology Confirmation	Treatment	Timing of Sampling	Technique	MiRNA Selection	Validation
Davey [14]	2022	Ireland	Blood	124	Prospective	Yes	NAC and surgery	First sample at breast cancer diagnosis, second sample halfway point during NAC	qRT-PCR	Selected from the literature	No
Zellinger [22]	2022	Austria	Tumour tissue	110	Retrospective	Yes	BCT, chemotherapy, anti-hormonal therapy, and radiotherapy	During BCT	qRT-PCR	miRNA array	Yes
Amiruddin [23]	2021	Indonesia	Plasma	34	Retrospective	Yes	Tamoxifen	1 year post-tamoxifen initiation	qRT-PCR	Selected from the literature	No
Thomopoulou [24]	2021	Greece	Plasma	204	Retrospective	Not mentioned	Surgery, anthracycline-based therapy, taxane-based therapy, anthracycline and taxane-based therapy	Post-surgery and pre-adjuvant therapy in those with early BC. Before initiation of first-line chemo in those with metastatic BC	qRT-PCR	miRNA array	No
Zellinger [25]	2020	Austria	Tumour tissue	81	Retrospective	Yes	Surgery, chemotherapy, anti-hormonal therapy, and radiotherapy	During surgery	qRT-PCR	Selected from the literature	Yes
Elango [26]	2020	Qatar	Tumour tissue	44	Retrospective	Yes	N/R	Lymph node metastasis and matched primary tumour tissue	qRT-PCR	miRNA array	No
Zhang [27]	2020	China	Tumour tissue	62	Retrospective	Yes	Surgery, chemotherapy, and hormone therapy	Not given	qRT-PCR	Selected from a literature review	No
Estevão-Pereira [28]	2019	Portugal	Tumour tissue and plasma	143 (98 tissue and 45 plasma)	Retrospective	Yes	N/R	Not given	qRT-PCR	Selected from historical laboratory findings	Yes
Giannoudis [29]	2019	UK	Tumour tissue	52	Retrospective	Yes	N/R	Not given	qRT-PCR	miRNA array	Yes
Masuda [30]	2018	Japan	Serum	366	Retrospective	Not mentioned	Surgery	Post-surgery	qRT-PCR	miRNA array	Yes
Wang [31]	2018	China	Tumour tissue	133	Retrospective	Yes	Surgery, endocrine therapy, radiation therapy, and chemotherapy	Post-surgery but pre-adjuvant therapy	qRT-PCR	Selected from a literature review	No
Bašová [32]	2017	Czech Republic	Serum	133	Retrospective	Yes	Surgery, anthracyclines, taxanes, trastuzumab, hormonal therapy, chemotherapy, and radiation	First—1 day pre-surgery Second—14–28 days after surgery Third—14–28 days after first non-surgical treatment Fourth—On relapse	qRT-PCR	Historical lab findings	No
Sueta [33]	2017	Japan	Tumour tissue and serum exosomes)	106 (32 from exosomes and 74 from tissue)	Retrospective	Not mentioned	Surgery, endocrine therapy, chemotherapy, and trastuzumab	Serum and tissue samples collected pre-surgery and pre-treatment.	qRT-PCR	miRNA array	No
Du [34]	2016	China	Tumour tissue	211	Retrospective	Not mentioned	Surgery followed by adjuvant chemotherapy or trastuzumab	During surgery	qRT-PCR	miRNA array	Yes
Huo [35]	2016	US	Serum	90	Retrospective	Yes	Surgery	One group pre-surgery, one group at time of recurrence, control pre-surgery	qRT-PCR	miRNA array	Yes
Sahlberg [36]	2015	Norway	Serum	194 (20 discovery and 110 validation)	Retrospective	Yes	Surgery and/or other unspecified treatment	Serum samples before surgery or any therapy	qRT-PCR	miRNA array	Yes
Marino [37]	2014	Brazil	Tumour tissue	64	Retrospective	Yes	N/R	Not given	qRT-PCR	miRNA array	No
Zhou [38]	2012	US	Tumour tissue	68 (16 screening, 52 validation)	Retrospective	Yes	Surgery, radiation therapy, and chemotherapy	Not given	qRT-PCR	miRNA array	Yes
Wu [39]	2012	US	Serum	68	Retrospective	Yes	NAC followed by surgery	Serum samples before neoadjuvant chemotherapy surgery	qRT-PCR	miRNA array	Yes

N; number, qRT-PCR: quantitative real-time polymerase chain reaction, UK; United Kingdom, US; United States, N/R; not reported, NAC; neoadjuvant chemotherapy, BCT; breast-conserving therapy.

### 3.4. Clinicopathological Data

The median age at diagnosis was 55.0 years (range: 27–85 years) (12 studies). Of the reported data, 76.7% of tumours were tumour stage 1-2 (1178/1536—12 studies), 42.0% had nodal involvement (382/910—6 studies), 36.8% were grade 3 (561/1,524—11 studies), and 87.0% had ductal histology (723/831—6 studies). Overall, 61.5% of patients were oestrogen-receptor-positive at the time of breast cancer diagnosis (1406/2287—19 studies) and 51.5% were progesterone-receptor-positive (1050/2037—15 studies). Furthermore, 31.5% of patients were HER2-positive (547/1738—13 studies). Overall, 42.0% of patients had nodal involvement (382/910—6 studies). Basic clinicopathological parameters are highlighted in Table 2.

### 3.5. MicroRNA Expression Profiling and Disease Recurrence

The 19 studies included in this review identified 44 miRNAs whose expression successfully predicted breast cancer recurrence. Results from nine studies assessed tumour miRNAs (47.4%), eight studies included circulatory miRNAs (42.1%), and two studies assessed both tumour and circulatory miRNAs (10.5%). Increased expression of 25 miRNAs and decreased expression of 14 miRNAs were exclusively identified in patients with recurrence compared to patients free of recurrence. Furthermore, five miRNAs (miR-17-5p, miR-93-5p, miR-130a-3p, miR-155, and miR-375) had conflicting data in the literature regarding their expression, with studies indicating both increased and reduced expression of these biomarkers impacting recurrence [24,31,32,33,35]. Of note, four studies used miRNA signatures to predict disease recurrence [24,29,34,35]. MicroRNA expression patterns in recurrence and their functional roles are shown in Table 3.

### 3.6. MicroRNA Expression from Tumour Tissue

Overall, 997 provided solid tumour tissue samples for evaluation (43.6%) as provided by 11 studies [22,25,26,27,28,29,31,33,34,37,38]. As mentioned, two of these eleven studies provided both tumour tissue and liquid biopsy [28,33]. Of these, seventeen miRNAs were upregulated in patients with recurrence and eight miRNAs were downregulated in patients with recurrence. There were six miRNAs which were assessed in both liquid biopsy and tumour tissue as discussed previously. Other than the contradictions already mentioned, there were no further contradictions in the findings of miRNA expression in tumour tissues and breast cancer recurrence.

### 3.7. MicroRNA Expression from Liquid Biopsy

Overall, 1290 provided liquid biopsy tissue (i.e., blood, serum, or plasma) for the evaluation of ct-miRNA (56.4%) as provided by 10 studies [14,23,24,28,30,32,33,35,36,39]. Two of these ten provided both tumour tissue and liquid biopsy [28,33]. Of the ten studies, three evaluated miRNAs expressed in the plasma, one measured miRNAs in the blood, and six measured miRNAs expressed in the serum. Of the microRNAs assessed in liquid biopsy, fifteen miRNAs were upregulated in patients with recurrence and ten miRNAs were downregulated in patients with recurrence. Six of these miRNAs (miR-17-5p, miR-30b-5p, miR-93-5p, miR-130a-3p, miR-340-5p, and miR-375) were assessed in both liquid biopsies as well as in tumour tissues. Of these six, four miRNAs (miR-17-5p, miR-93-5p, miR-130a-3p, and miR-375) had contradicting findings where some studies correlated over expression with recurrence and other studies correlated under expression with recurrence [31,33,35]. Furthermore, one miRNA (miR-155) assessed in liquid biopsy in two separate studies showed contradicting findings [24,32]. MicroRNA relapse patterns in liquid biopsies and tumour tissues are displayed in Table 4.

### 3.8. Timepoints of Tissue Extraction

Of the nineteen included studies, eleven studies obtained patient tissue at one timepoint during their study [22,23,24,25,26,30,31,33,34,36,39], while five studies did not report the timing of when their samples were taken [27,28,29,37,38]. Three studies measured miRNA expression at varying timepoints (i.e., at diagnosis, pre-surgery, post-surgery, pre-adjuvant therapy, and post-adjuvant therapy) [14,32,35] (Figure 2).

## 4. Discussion

This study is the first systematic review assessing the roles of various miRNAs in predicting breast cancer recurrence. This work illustrates that miRNAs may have potential utility in enhancing the prognostication for prospective patients. In total, this study successfully identified 44 miRNAs which were previously shown to predict breast cancer recurrence, with several unique targets distinguishing those likely to suffer LRR and DR at long-term follow up. These are important results, which may be pragmatically included in target panels for future clinical research trials, which ultimately are necessitated to elicit the true value of these biomarkers in estimating recurrence in breast cancer.

In this review, four studies which had developed a miRNA signature that may be useful in predicting disease recurrence were included: Thomopoulou et al. derived a 4-miRNA panel consisting of miR-19-a, miR-20-a, miR-126, and miR-155 which aimed to distinguish early from metastatic breast cancer patients (combined predictive ability: 80.2%, sensitivity: 75%, specificity: 76%) [29]. Additionally, the work of Giannoudis et al. aimed to derive a miRNA signature that could predict brain metastasis in patients with breast cancer and found 11 miRNA targets of interest. Following validation, a 4-miRNA signature consisting of miR-132-3p, miR-199a-5p, miR-150-5p, and miR-155-5p identified patients with brain metastases with a predictive ability of 82% (sensitivity: 76.7%, specificity: 83.3%) [30]. Du et al. discovered and validated a 2-miRNA signature consisting of miR-150, which is highly expressed in the serum of patients with recurrence, and miR-4734, which is under expressed in the serum of patients with recurrence (reported predictive ability of 69%, 73%, and 67% in the training, internal, and external independent validation set, respectively). Interestingly, these authors went further on to combine their 2-miRNA signature with TNM stage and found a superior and significant predictive ability of 71% compared to 67% in the external independent set of the signature [31]. Huo et al. analysed miRNAs individually in their analysis of serum samples and identified that a high expression of miR-194-5p and a low expression of miR-375 correlated with worse breast cancer prognosis. However, when pooling five different miRNAs (miR-21-5p, miR-205-5p, miR-382-5p, miR-376c-3p, and miR-411-5p) in patients with the triple-negative biomolecular subtype, it gave a more accurate miRNA signature (predictive ability of 81% vs. 0% in breast cancer patients with early recurrence versus breast cancer patients without recurrence) [32]. As illustrated by these authors, and given the non-specific characteristics of miRNAs, it seems likely that combining several targets may be the most pragmatic solution to enhancing the diagnostic accuracy of miRNA profiling in detecting disease recurrence.

This review has further highlighted the multifunctional and non-specific role of miRNAs in oncogenesis. For example, these data outlined the different miRNAs which were found to be relevant in predicting breast cancer recurrence among studies, while also highlighting discordant information surrounding miRNA expression profiles and recurrence. For example, Thomopoulou et al. outlined how decreased expression of plasma miR-155 was associated with breast cancer recurrence [24], while increased expression of this biomarker was associated with recurrence in two other studies [29,32]. Moreover, while Davey et al. evaluated the relevance of miR-155 in breast cancer, it failed to correlate with recurrence in their study [14]. Wang et al. illustrated increased expression of miR-17-5p in the serum of patients with breast cancer recurrence [31], while the study by Sueta et al. found reduced expression of exosomal miR-17-5p to be associated with disease recurrence [33]. Furthermore, discrepancies were found by Sueta et al. between three miRNAs expressed in tumour tissues with the same miRNAs expressed in the serum from the same patient cohort: Three miRNAs (miR-17-5p, miR-93-5p, and miR-130a-3p) were found to be overexpressed in tumour tissues of patients with recurrence while the same miRNAs were found to be downregulated in the serum of patients with recurrence [33]. As previously alluded to, this highlights the ubiquitous and multifunctional nature of miRNAs in human biology.

In addition, this review emphasizes the importance of performing validation of the results pertaining to miRNA profiling, with just 10 of the 19 studies included validating their findings in an independent cohort of patients following the identification of their targets in a discovery cohort [22,25,28,29,30,34,35,36,38,39]. For example, Wu et al. found reduced expression of miR-375 was associated with breast cancer recurrence in the discovery cohort; however, they failed to validate these findings in an independent patient cohort [39]. Moreover, Huo et al. found that increased expression of miR-375 was predictive of patients likely to suffer breast cancer recurrence, and these findings were validated internally [35]. In the absence of an expert consensus recommendation to appropriately appraise miRNA profiling, translational research efforts may remain unvalidated and therefore negatively impacting the translatability of such results into clinical practice.

This systematic review is subject to several limitations. Firstly, 18 of the 19 studies included in this review were of retrospective design, rendering the evidence subject to inherent biases, including selection and confounding biases. These factors inevitably impact negatively upon the reliability and transferability of results from the bench to the bedside. In addition, the optimal medium from which the samples were obtained and analysed from (i.e., plasma, serum, whole blood, or tumour tissue) was often left to the discretion of the clinician or scientist with no consensus available, further placing scrutiny under the consistency of the results yielded. A strong point of this study was the evidence of solid consensus surrounding the miRNA profiling methodology where all 19 included studies utilised real-time quantitative reverse transcription polymerase chain which supports the consistency of the results [14,22,23,24,25,26,27,28,29,30,31,32,33,34,35,36,37,38,39]. However, this technique itself is challenging as most miRNAs will require extensive cycles of amplification, and absolute quantification remains difficult.

Two important points to further discuss include the lack of prospective trials included in this review, along with a scarcity of a coherent approach among the studies. The need for prospective trials will allow tailoring of the studies to specifically target a certain miRNA and adequately assess its role in recurrence. The exciting potential of miRNAs in oncogenesis and the recent findings in the field of translational research calls for the application of these retrospective studies into modern trials. In addition, there is a lack of coherence in these studies. For example, styling of these studies according to molecular subtypes may give exciting results which will allow for potentially exciting advancements. Only six of the nineteen included studies carried out their analysis on a specific molecular subtype group [23,24,27,34,36,38]. The heterogeneity and relative lack of coherence in the patient groups are outlined in Table 2.

There were several mechanistical processes which connected the miRNAs with relation to breast cancer recurrence. For example, six miRNAs identified (miR-103, miR-107, miR-205-5p, miR-214-3p, miR-30b-5p, miR375) were involved in epithelial–mesenchymal transition. These links in mechanistical processes may direct future prospective trials to further assess them in breast cancer patients and evaluate their clinical significance. In addition, a significant link to address is between miRNAs and specific molecular subtypes of breast cancer. There is research suggesting a link between the role a certain microRNA may play and the subtype of breast cancer, as expression levels and functions of miRNAs differ among the various subtypes of breast cancer [40]. Different subtypes of breast cancer hold fundamental differences in incidence, treatment response, disease progression, survival rates, and imaging [41]. Among the studies included in this review, only five papers mentioned the subtype of breast cancer, yet none discussed the actual mechanistic link between them [23,27,34,36,38]. Larger clinical cohort studies and prospective trials are needed to verify the direct link between how an mRNA is regulated and linked to breast cancer recurrence to the molecular subtype of breast cancer (such as HER2+, luminal, etc.). We provide our summative conclusions on this topic as a drive to further direct research in this area. It must be noted that in addition to their role in predicting the outcome of breast cancer, miRNAs are also known to predict response to chemotherapy (chemoresistance). One example is miRNA-195 which was found to induce resistance to chemotherapy in breast cancer by reducing the levels of a protein known as SEMA6D [42].

In conclusion, this study is the first systematic review to illustrate the roles of miRNAs in predicting long-term recurrence for primary breast cancer. A miRNA panel was successfully identified which may be useful in predicting disease recurrence, and may have potential applications in clinical practice to improve patient-specific prognostication. Notwithstanding, the provision of well-designed prospective, multicentre clinical trials, such as the ICORG 10/11 trial, are required to further validate such findings prior to application to clinical practice. There is also a need for a greater number of prospective studies in this field and a more coherent approach to optimise utility.

## Figures and Tables

**Figure 1 ijms-24-07115-f001:**
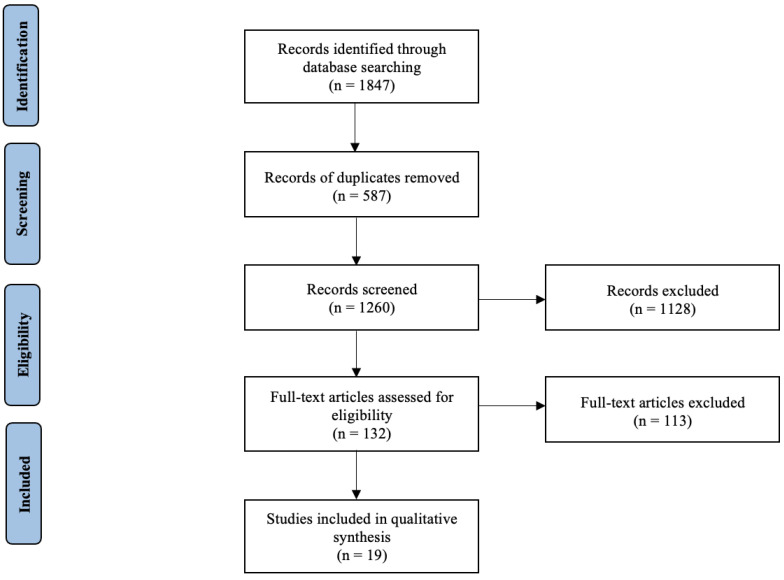
PRISMA flow diagram detailing the systematic search process.

**Figure 2 ijms-24-07115-f002:**
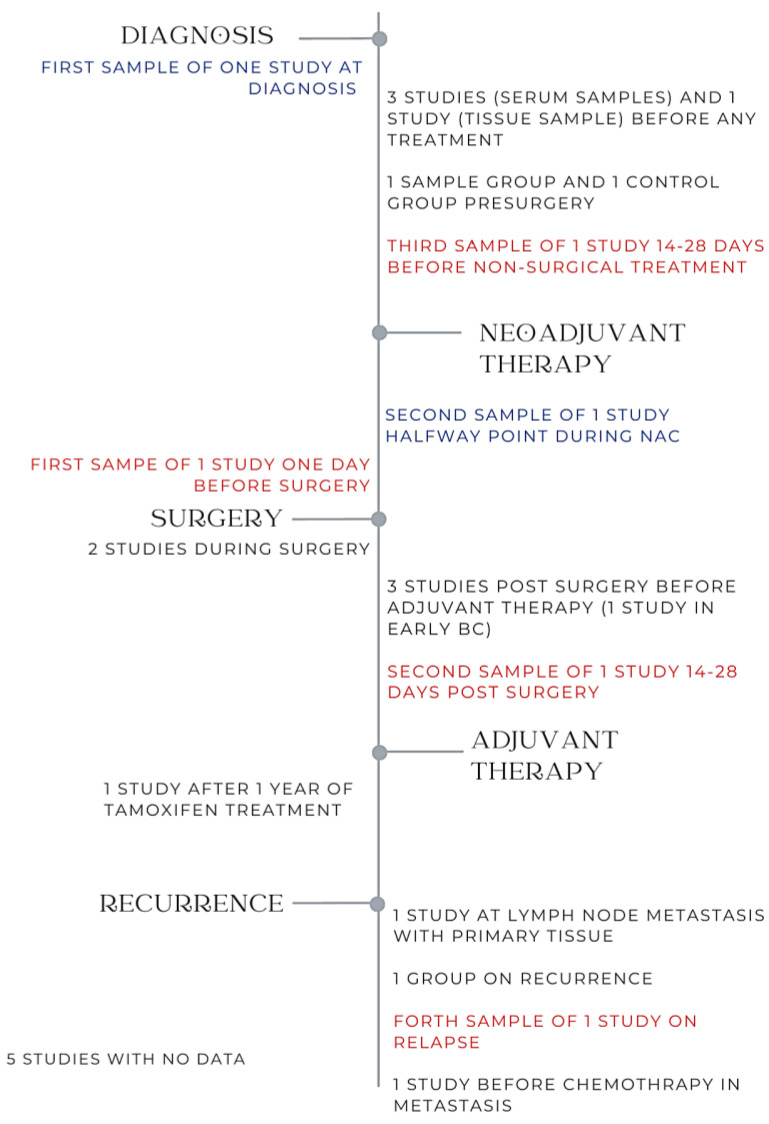
Timepoints at which liquid and solid tumour biopsies were obtained.

**Table 2 ijms-24-07115-t002:** Basic clinicopathological parameters from the included studies.

Author	Median Age	Over 50	Age Range	Relapse	ER+	PR+	HER2+	TNBC	G1	G2	G3	S1	S2	S3	S4	LRR	DR	Nodal Involvement	Ductal Histology
Davey [14]	55.0	N/R	48–63	N/R	81	66	38	25	1	66	57	N/R	N/R	N/R	N/R	N/R	N/R	79	N/R
Zellinger [22]	53.0	N/R	33–79	37	81	74	38	N/R	1	68	34	85	25	0	0	23	0	N/R	89
Amiruddin [23]	N/R	N/R	N/R	15	34	19	N/R	N/R	0	12	11	N/R	N/R	N/R	N/R	N/R	N/R	N/R	N/R
Thomopoulou [24]	56.0	N/R	27–82	121	138	133	235	31	32	56	107	122	103	10	0	N/R	N/R	N/R	N/R
Zellinger [25]	55.0	N/R	40–79	27	54	38	35	0	4	45	32	61	20	0	0	N/R	N/R	N/R	N/R
Elango [26]	52.0	N/R	32–74	44	36	33	10	4	N/R	N/R	N/R	0	25	14	5	N/R	N/R	N/R	N/R
Zhang [27]	N/R	19	N/R	28	62	N/R	N/R	0	N/R	N/R	N/R	30	32	0	0	N/R	N/R	N/R	N/R
Estevão-Pereira [28]	57.0	N/R	28–82	140	131	109	33	6	10	75	56	47	72	11	11	N/R	N/R	N/R	N/R
Giannoudis [29]	N/R	N/R	N/R	40	30	N/R	14	8	N/R	N/R	N/R	N/R	N/R	N/R	N/R	0	40	N/R	N/R
Masuda [30]	N/R	N/R	N/R	125	243	192	100	34	214		98	173	153		N/R	N/R	N/R	120	356
Wang [31]	54.9	N/R	N/R	29	76	68	16	0	58		75	108	108	25		N/R	N/R	72	N/R
Bašová [32]	61.5	N/R	37–84	13	119		4	12	27	72	27	94	40	0	0	N/R	N/R	26	97
Sueta [33]	55.5	N/R	30–79	16	19	16	N/R	6	N/R	19	13	10	17	2	N/R	N/R	N/R	N/R	N/R
Du [34]	N/R	79	N/R	49	124		211	N/R	N/R	N/R	N/R	N/R	N/R	N/R	N/R	N/R	N/R	N/R	N/R
Huo [35]	50.0	N/R	N/R	28	41	38	11	40	5	30	51	40	29	13	5	8	20	46	76
Sahlberg [36]	N/R	N/R	N/R	10	33	0	0	130	N/R	N/R	N/R	N/R	N/R	N/R	N/R	N/R	N/R	N/R	N/R
Marino [37]	53.1	N/R	29–95	29	34	21	13	0	N/R	N/R	N/R	27	18	9	10	0	29	39	49
Zhou [38]	52.3	N/R	N/R	23	49	N/R	N/R	N/R	N/R	N/R	N/R	N/R	N/R	N/R	N/R	23	0	N/R	56
Wu [39]	N/R	N/R	N/R	19	21	N/R	N/R	N/R	N/R	N/R	N/R	N/R	N/R	N/R	N/R	N/R	N/R	N/R	N/R

N; number, N/R; not reported, ER; oestrogen receptor, PR; progesterone receptor, HER2; human epidermal growth factor receptor-2, TNBC; triple-negative breast cancer, G; histological grade, S; tumour stage, LRR; loco-regional recurrence, DR; distant recurrence.

**Table 3 ijms-24-07115-t003:** MicroRNA expression patterns in recurrence and their functional roles.

miRNA	Expression	Role	Ref
miR-9	High expression in patients with recurrence	Targets E-cadherin, facilitating metastases and stimulating angiogenesis in BC	[38]
Early breast cancer		miR-10b: inhibits natural killer cells to recognize and attack cancer cells miR-155: regulates the differentiation of B lymphocytes and CD4+ T lymphocytes and the activation of T regulator cells miR-126: reshapes tumour microenvironment and represses recruitment of mesenchymal cells and inflammatory monocytes	[24]
miR-126, miR-155, and miR10b	Low expression in patients with recurrence		
miR-19a	High expression in patients with recurrence		
number of infiltrated axillary lymph nodes and low miR-10b expression	Independent predictor for short disease-free survival		
axillary lymph nodes and low miR-126	Shorter overall survival		
Metastatic breast cancer			
recurrent disease and low miR-10b expression	Short PFS	miR-19a: oncogenic role by contributing to shifting of the M2 to M1-like phenotype of TAMs	
miR-17-5p	High expression in patients with recurrence	Involved in tumour proliferation through the modulation of the PI3K/Akt/mTOR pathway	[31]
miR-18b, miR-103, miR-107, and miR-652	High expression in TNBC patients with recurrence	miR-103 and miR-107: play a role in EMT and DNA repair miR-652: unclear oncogenic role in BC miR-18b: promotes BC cell migration and metastasis	[36]
miR-194-5p	High expression in serum of patients with recurrence through downregulation of TSGs	miR-194-5p miR-21-5p: increases cell growth, invasion, and migration, and reduces apoptosis	[35]
miR-375	Low expression in patients with recurrence	miR-375: involved in EMT	
Seven miRNA signatures (miR-21-5p, miR-375, miR-205-5p, miR-194-5p, miR-382-5p, miR-376c-3p, and miR-411-5p)	Individually, only the two miRNAs mentioned above are significant, but when pooled together, they provide the best predictive value: High expression of miR-21-5p, miR-375, miR-205-5p, and miR-194-5p associated with recurrence; Low expression of miR-382-5p, miR-376c-3p, and miR-411-5p associated with recurrence.	miR-205-5p, miR-382-5p, miR-376c-3p, and miR-411-5p: unclear role in BC	
miR-155 and miR-24	High expression in patients with recurrence	Modulate the TGFβ pathway albeit by targeting different mechanisms	[33]
miR-30b-5p	High expression in patients with recurrence	Oncogenic role; facilitates EMT	[28]
miR-340-5p, miR-17-5p, miR-130a-3p, and miR-93-5p	High expression in patients with recurrence	miR-340-5p and miR-130a-3p: involved in tumour proliferation	[32]
miR-17-5p, miR-130a-3p, and miR- 93-5p	Low expression in patients with recurrence	miR-17-5p: involved in tumour proliferation	
miR-340-5p	High expression in patients with recurrence	miR-93-5p: role remains unclear	
miR-122	High expression in patients with recurrence	Unclear role in BC	[39]
miR-132-3p	Low expression in patients with brain metastasis	Interfere with inflammatory networks involved in metastasis	[29]
miR-199a-5p, miR-150-5p, and miR-155-5p	High expression in patients with brain metastasis		
miR-145	Low expression in patients with recurrence; High expression correlated with improved RFS and DFS	Acts as a tumour suppressor	[14]
miR-150	High expression in tumour tissue of patients with recurrence; worse DFS	miR-150: involved in cell proliferation	[34]
miR-4734	Low expression in tumour tissue of patients with recurrence; worse DFS	miR-4734:	
miR-183, miR-494, and miR-21	High expression in patients with recurrence	miR-183: associated with migration and invasion miR-949 and miR-21: target cancer-related molecules including PTEN	[37]
miR-205 and miR-214	Low expression in patients with recurrence and poor overall survival	miR-205 and miR-214: suppress BC cell proliferation, migration and colony formation miR-200 family: regulate BMI1 expression in BC tumour-initiating cells and suppress EMT	[26]
miR-200 family (miR-200a, miR-200b, and miR-200c)	Low expression linked to lymph node metastasis		
miR-205-5p and miR-214-3p	Together, high expression of these two markers in lymph node metastasis		
miR-221	High levels in patients with local recurrence and metastasis High levels in patients with tamoxifen resistance	Alteration of cell cycle processes and evasion of apoptosis	[23]
miR-375	Low expression in patients with recurrence	Inhibitory function on cell proliferation	[25]
miR-891a-5p and miR-383-5p	Low expression in patients with recurrence	miR-891a-5p: inhibits BC T47D and MCF7 cell proliferation and metastasis miR-383-5p: suppresses cancer cell proliferation and migration	[27]
miR-488	High expression in serum of patients with recurrence	Unexpectedly associated with worse prognosis in HER2+ BC, despite its role as a tumour suppressor	[30]
miR-3651	High expression in patients with recurrence	Interferes with protein binding as well as cytoskeletal and cell membrane stability	[22]

RFS; recurrence-free survival, DFS: disease-free survival, BC; breast cancer, EMT; epithelial–mesenchymal transition.

**Table 4 ijms-24-07115-t004:** MicroRNA relapse patterns in liquid biopsies and tumour tissues.

**Solid tumour tissue MicroRNA for relapse prediction**					
**Author**	**Year**	**miRNAs**	**Tissue**	**Molecular Subtype**	**Relapse**
Zellinger [23]	2022	miR-3651	Tumour tissue	Not specified	Local and distant
Zellinger [25]	2020	miR-375	Tumour tissue	Not specified	Local
Elango [26]	2020	miR-205-5p, miR-214-3p, miR-200	Tumour tissue	Not specified	Local
Zhang [27]	2020	miR-891a-5p and miR-383-5p	Tumour tissue	Luminal	Distant
Estevão-Pereira [28]	2019	miR-30b-5p	Tumour tissue	Not specified	Local and distant
Giannoudis [29]	2019	miR-132-3p, miR-199a-5p, miR-150-5p, and miR-155-5p	Tumour tissue	Not specified	Distant (brain)
Wang [31]	2018	miR-17-5p	Tumour tissue	Not specified	Not specified
Sueta [33]	2017	miR-340-5p, miR-17-5p, miR-130a-3p, and miR-93-5p	Tumour tissue	Not specified	Not specified
Du [34]	2016	miR-150, miR-4734	Tumour tissue	HER2+	Not specified
Marino [37]	2014	miR-183, miR-494, and miR-21	Tumour tissue	Not specified	Distant
Zhou [38]	2012	miR-9	Tumour tissue	Luminal	Local
**Liquid biopsy MicroRNA for relapse prediction**					
**Author**	**Year**	**miRNAs**	**Tissue**	**Molecular Subtype**	**Relapse**
Davey [14]	2022	miR-145	Blood	Not specified	Not specified
Amiruddin [23]	2021	miR-221	Plasma	Luminal	Local and distant
Thomopoulou [24]	2021	miR-155, miR-10b, miR-126, miR-19a	Plasma	Not specified	Local
Estevão-Pereira [28]	2019	miR-30b-5p	Plasma	Not specified	Local and distant
Masuda [30]	2018	miR-488	Serum	Not specified	Local
Bašová [32]	2017	miR-155 and miR-24	Serum	Not specified	Not specified
Sueta [33]	2017	miR-340-5p, miR-17-5p, miR-130a-3p, and miR-93-5p	Serum exosomes	Not specified	Not specified
Huo [35]	2016	miR-194-5p miR-375 Seven miRNA signatures (miR-21-5p, miR-375, miR-205-5p, miR-194-5p, miR-382-5p, miR-376c-3p, and miR-411-5p)	Serum	Not specified	Local and distant
Sahlberg [36]	2015	miR-18b, miR-103, miR-107, and miR-652	Serum	TNBC	Distant
Wu [39]	2012	miR-122	Serum	Not specified	Local

## Data Availability

Not applicable.

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
