# Peer review of "Assessing the Role of MicroRNAs in Predicting Breast Cancer Recurrence—A Systematic Review"

_ijms, 2023, doi:10.3390/ijms24087115_

Round 1

Reviewer 1 Report

The authors have summarized an important topic on the miRNA expression patterns and its capability to predict breast cancers. This data indeed has a great potential and applicability to identify patients with probable breast cancer recurrence and an approach to improve the patient survival rates upon neoadjuvant therapy or surgery.

miRNAs in cancer do not only play role in cancer recurrence directly but also assist cancer cells to develop resistance and indirectly results in the recurrence of drug resistant cancers.

In line 20, ‘Mi(cro)RNA’ can be correctly written as miRNA.

The authors could include the miRNA 195 as it plays several role in cancer recurrence, drug resistance and cancer progression. (https://doi.org/10.3390/cancers13235979).

Author Response

Reviewer #1

Reviewer’s Comment:

The authors have summarized an important topic on the miRNA expression patterns and its capability to predict breast cancers. This data indeed has a great potential and applicability to identify patients with probable breast cancer recurrence and an approach to improve the patient survival rates upon neoadjuvant therapy or surgery.

miRNAs in cancer do not only play role in cancer recurrence directly but also assist cancer cells to develop resistance and indirectly results in the recurrence of drug resistant cancers.

My comments are below.

Author’s Response:

Thank you for taking the time to review our manuscript which is being considered for publication in the International Journal of Molecular Sciences.

We believe the manuscript has benefitted greatly from this review.

Reviewer’s Comment:
1. In line 20, ‘Mi(cro)RNA’ can be correctly written as miRNA.

Author’s Response:

Thank you for this suggestion.

We have made corrections accordingly.

‘MiRNAs are small, non-coding RNA molecules which are known to regulate genetic expression and have previously demonstrated relevance as biomarkers in malignancy.

Reviewer’s Comment:

  1. The authors could include the miRNA 195 as it plays several role in cancer recurrence, drug resistance and cancer progression. (https://doi.org/10.3390/cancers13235979).

Author’s Response:

Thank you for this comment.

We have added a perspective section at the end of discussion which incorporates the significant role of miRNA-195 in breast cancer:

It must be noted that in addition to their role in predicting the outcome of breast cancer, miRNAs are also known to predict response to chemotherapy (chemoresistance). One example is miRNA-195 which was found to induce resistance to chemotherapy in breast cancer by reducing the levels of a protein known as SEMA6D (42)

Thank you for providing a thorough review of our manuscript. 

Reviewer 2 Report

In this review article the authors methodically searched and analyzed, summarized, and categorically presented 19 primary research articles concerning predicting value of microRNA expression in breast cancer recurrence. This is an important area of cancer research, in general, and the review has potential utility for informed study design in prospective clinical studies or addressing basic science questions in breast cancer recurrence and metastases. However, addition of the following aspect will increase the persuasiveness of the review. Beside discussing each of the studies in the Discussion section, authors should include a Perspective section to elaborate their view on any overlapping mechanistic pathways or biochemical processes that might connect the 44 miRNA with breast cancer recurrence, and how these 44 miRNA could be adopted in larger clinical cohort or preclinical studies, and what are the open questions needs to be address in future studies.

Additionally, few minor points should be addressed:

Line 98: What were the molecular subtypes/HR status of recurred tumors for corresponding patients? How many of these studies reports systemic and local lymph node metastases?

Line 89, 94: Median time is more relevant for patient follow up timeline or age reporting. Please include median values.

Author Response

Reviewer #2

Reviewer’s Comment:

In this review article the authors methodically searched and analyzed, summarized, and categorically presented 19 primary research articles concerning predicting value of microRNA expression in breast cancer recurrence. This is an important area of cancer research, in general, and the review has potential utility for informed study design in prospective clinical studies or addressing basic science questions in breast cancer recurrence and metastases.

Author’s Response:

Thank you for taking the time out of your busy schedule to review our manuscript. The manuscript has benefitted greatly from this thorough review.

Reviewer’s Comment:
However, addition of the following aspect will increase the persuasiveness of the review. Beside discussing each of the studies in the Discussion section, authors should include a Perspective section to elaborate their view on any overlapping mechanistic pathways or biochemical processes that might connect the 44 miRNA with breast cancer recurrence, and how these 44 miRNA could be adopted in larger clinical cohort or preclinical studies, and what are the open questions needs to be address in future studies.

Author’s Response:

Thank you for this valuable suggestion. We have added a perspective section at the end of the discussion as per your comment:

‘There were several mechanistical processes which connected the miRNAs with relation to breast cancer recurrence. For example, 6 miRNAs identified (miR-103, miR-107, miR-205-5p, miR-214-3p, miR-30b-5p, miR375) were involved in epithelial-mesenchymal transition. These links in mechanistical processes may direct future prospective trials to further assess them in breast cancer patients and evaluate their clinical significance. In addition, a significant link to address is between miRNAs and specific molecular subtypes of breast cancer. There is research suggesting a link between the role a certain microRNA may play and the subtype of breast cancer, as expression levels and functions of miRNA differ among the various subtypes of breast cancer (40). Different subtypes of breast cancer hold fundamental differences in incidence, treatment response, disease progression, survival rates, and imaging (41). Among the studies included in this review, only five papers mentioned the subtype of breast cancer, yet none discussed the actual mechanistic link between them (23, 27, 34, 36, 38). Larger clinical cohort studies and perspective trials are needed to verify the direct link between how a mRNA is regulated and linked to breast cancer recurrence to the molecular subtype of breast cancer (such as HER2+, luminal, etc.). We provide our summative conclusions on this topic as a drive to further direct research in this area. It must be noted that in addition to their role in predicting the outcome of breast cancer, miRNAs are also known to predict response to chemotherapy (chemoresistance). One example is miRNA-195 which was found to induce resistance to chemotherapy in breast cancer by reducing the levels of a protein known as SEMA6D (42).’

Reviewer’s Comment:

Additionally, few minor points should be addressed:

Line 98: What were the molecular subtypes/HR status of recurred tumors for corresponding patients? How many of these studies reports systemic and local lymph node metastases?

Author’s Response:

Thank you for this comment.

In relation to the molecular subtypes, the molecular subtypes of the patients included in these studies have been identified in Table 2. Lymph node involvement was included in Table 2 and a further line has been added to the ‘Clinicopathological Data’ section as follows:

‘Overall, 42.0% of patients had lymph node involvement (382/910 – 6 studies).’

Reviewer’s Comment:
Line 89, 94: Median time is more relevant for patient follow up timeline or age reporting. Please include median values.

Author’s Response:

Thank you for this suggestion.

The median age of patients at diagnosis and the median follow up have now replaced mean values. Amendments have also been made to Table 2, accordingly.  

‘In total, 34.7% of patients experienced disease recurrence at median follow up of 78.9 months (793/2,287 – 18 studies).’

‘The median age at diagnosis was 55.0 years (range: 27 – 85 years) (12 studies).’

Thank you for this thorough review of our manuscript.

Reviewer 3 Report

Authors performed a systematic review trying to evaluate the role of miRNAs in predicting breast cancer recurrence, and they concluded that MiRNA expression patterns have the ability to predict breast cancer recurrence, and their findings may be used in future translational research studies to identify patients with breast cancer recurrence to improve oncological and survival outcomes for our prospective patients. Even though the function of miRNA to act as a predictor of cancer recurrence is not a novel topic which has been studied for years. The intention of this study is meaningful. However, apparent flaws also exist. This manuscript should be considered for acceptance after serious revision.

1.As we all know, there are different subtypes of breast cancer: 1.luminal A, 2.luminal B,3.human epidermal growth factor receptor 2 (HER2)-enriched, and 4.basal-like. More importantly, different subtypes of breast cancer hold critical differences in incidence, response to treatment, disease progression, survival, and imaging features(doi.org/10.1093/jbi/wbaa110).In addition, as mentioned by Sasagu K’s team that “the expression levels and functions of miRNA differ among the various subtypes of breast cancer, and it is necessary to take account of the characteristics of each breast cancer subtype during research into the roles of miRNA in breast cancer”(DOI: 10.1038/jhg.2016.89).The authors also mentioned in line 164 “this highlights the ubiquitous and multifunctional nature of miRNAs in human biology”. Therefore, the miRNAs selected in this manuscript should be analyzed based on different subtypes of breast cancer, not as a whole one. Current results retrieved from data of unclassified breast cancer hard to provide precise conclusion.

2. Please claim that whether the primary and recurrence breast cancers included in this study were confirmed by pathology.

3. The information of the methods applied for diagnosis should be added in Table 1.

4. The information of pathological types of breast cancer should be improved in Table 2.

Author Response

Reviewer #3

Reviewer’s Comment:
Authors performed a systematic review trying to evaluate the role of miRNAs in predicting breast cancer recurrence, and they concluded that MiRNA expression patterns have the ability to predict breast cancer recurrence, and their findings may be used in future translational research studies to identify patients with breast cancer recurrence to improve oncological and survival outcomes for our prospective patients. Even though the function of miRNA to act as a predictor of cancer recurrence is not a novel topic which has been studied for years. The intention of this study is meaningful. However, apparent flaws also exist. This manuscript should be considered for acceptance after serious revision.

Author’s Response:

Thank you for taking the time to review our manuscript which is being considered for publication in the International Journal of Molecular Sciences.

Reviewer’s Comment:
1.As we all know, there are different subtypes of breast cancer: 1.luminal A, 2.luminal B,3.human epidermal growth factor receptor 2 (HER2)-enriched, and 4.basal-like. More importantly, different subtypes of breast cancer hold critical differences in incidence, response to treatment, disease progression, survival, and imaging features(doi.org/10.1093/jbi/wbaa110).In addition, as mentioned by Sasagu K’s team that “the expression levels and functions of miRNA differ among the various subtypes of breast cancer, and it is necessary to take account of the characteristics of each breast cancer subtype during research into the roles of miRNA in breast cancer”(DOI: 10.1038/jhg.2016.89).The authors also mentioned in line 164 “this highlights the ubiquitous and multifunctional nature of miRNAs in human biology”. Therefore, the miRNAs selected in this manuscript should be analyzed based on different subtypes of breast cancer, not as a whole one. Current results retrieved from data of unclassified breast cancer hard to provide precise conclusion.

Author’s Response:

Thank you for this comment.

We agree with your comment regarding the importance of analysing miRNAs based on the different subtypes of breast cancer. As per your comment, we have added a new column in Table 4 titled ‘molecular subtypes’. Unfortunately, only 5 of the 19 included studies assessed miRNAs in a specific subtype of breast cancer and not a single study found a significant link of the identified miRNAs with a specific subtype of breast cancer. This does call for further research and prospective trials which will help enhance the utility of miRNAs, and this has now been included in a Perspective section in our discussion to highlight its importance.

In addition, a significant link to address is between miRNAs and specific molecular subtypes of breast cancer. There is research suggesting a link between the role a certain microRNA may play and the subtype of breast cancer, as expression levels and functions of miRNA differ among the various subtypes of breast cancer (40). Different subtypes of breast cancer hold fundamental differences in incidence, treatment response, disease progression, survival rates, and imaging (41). Among the studies included in this review, only five papers mentioned the subtype of breast cancer, yet none discussed the actual mechanistic link between them (23, 27, 34, 36, 38). Larger clinical cohort studies and perspective trials are needed to verify the direct link between how a mRNA is regulated and linked to breast cancer recurrence to the molecular subtype of breast cancer (such as HER2+, luminal, etc.). We provide our summative conclusions on this topic as a drive to further direct research in this area.’

Reviewer’s Comment:
2. Please claim that whether the primary and recurrence breast cancers included in this study were confirmed by pathology.

Author’s Response:

Thank you for this suggestion.

This has now been added to Table 1 and mentioned under ‘Disease Recurrence’.

Pathological confirmation of primary and recurrent breast cancer was specifically included in 15 studies (14, 22-23, 25-29, 31-32, 35-39).

Reviewer’s Comment:
3. The information of the methods applied for diagnosis should be added in Table 1.

Author’s Response:

Thank you for this suggestion.

We have reviewed the studies assessing for methods of diagnosis mentioned and we came to a conclusion that the vast majority of studies do not discuss the methods applied and we feel that this column would not be of relevance to the aim of our review.

Reviewer’s Comment:
4. The information of pathological types of breast cancer should be improved in Table 2.

Author’s Response:

Thank you for this suggestion.

We found most studies classified their patient to include hormone receptor status rather than Luminal or Luminal B in terms of molecular subtyping. Therefore, we concluded that displaying the pathological types in the way we did is the most efficient.

Thank you for taking the time out from your busy schedule to review our manuscript.

Round 2

Reviewer 3 Report

    Authors performed a systematic review trying to evaluate the role of miRNAs in predicting breast cancer recurrence, and they concluded that MiRNA expression patterns have the ability to predict breast cancer recurrence, and their findings may be used in future translational research studies to identify patients with breast cancer recurrence to improve oncological and survival outcomes for our prospective patients. Even though the function of miRNA to act as a predictor of cancer recurrence is not a novel topic which has been studied for years. This Systematic Review still possesses certain positive effect in such field.